# Oligodendroglial Energy Metabolism and (re)Myelination

**DOI:** 10.3390/life11030238

**Published:** 2021-03-13

**Authors:** Vanja Tepavčević

**Affiliations:** Achucarro Basque Center for Neuroscience, University of the Basque Country, Parque Cientifico de la UPV/EHU, Barrio Sarriena s/n, Edificio Sede, Planta 3, 48940 Leioa, Spain; vanja.tepavcevic@achucarro.org

**Keywords:** energy metabolism, oligodendrocyte, oligodendrocyte progenitor cell, myelin, remyelination, multiple sclerosis, glucose, ketone bodies, lactate, N-acetyl aspartate

## Abstract

Central nervous system (CNS) myelin has a crucial role in accelerating the propagation of action potentials and providing trophic support to the axons. Defective myelination and lack of myelin regeneration following demyelination can both lead to axonal pathology and neurodegeneration. Energy deficit has been evoked as an important contributor to various CNS disorders, including multiple sclerosis (MS). Thus, dysregulation of energy homeostasis in oligodendroglia may be an important contributor to myelin dysfunction and lack of repair observed in the disease. This article will focus on energy metabolism pathways in oligodendroglial cells and highlight differences dependent on the maturation stage of the cell. In addition, it will emphasize that the use of alternative energy sources by oligodendroglia may be required to save glucose for functions that cannot be fulfilled by other metabolites, thus ensuring sufficient energy input for both myelin synthesis and trophic support to the axons. Finally, it will point out that neuropathological findings in a subtype of MS lesions likely reflect defective oligodendroglial energy homeostasis in the disease.

## 1. Introduction

Myelination is the key evolutionary event in the development of higher vertebrates. This process significantly accelerates the propagation of action potentials without increasing axonal diameter [1]. In addition, myelin has a neurotrophic role both in the central nervous system (CNS) and the peripheral nervous system (PNS) [2]. Importantly, both myelin loss and alterations in the composition of myelin compromise axonal health and can lead to neurodegeneration [3,4,5].

Myelin is a multilamellar sheath that consists of water and dry weight. The dry weight consists predominantly of lipids (70–80%) and, to a lesser extent, proteins (20–30%) [6].

In the CNS, myelin is produced by oligodendrocytes. Unlike Schwann cells, the myelinating cells of the PNS, oligodendrocytes produce up to 50–80 internodes on multiple axons (reviewed in [7]), which means that alterations in a small number of these cells affect a significant number of axons. Importantly, alterations of oligodendrocytes and CNS myelin are a feature of several CNS pathologies including leukodystrophies (genetic disorders that lead to dys/de-myelination), spinal cord injury (in which myelin loss around spared axons significantly contributes to permanent damage), multiple sclerosis (MS; a chronic inflammatory demyelinating disease that leads to neurodegeneration) and other inflammatory demyelinating CNS disorders, periventricular leukomalacia (white matter disease of the newborn), as well as several “classical” neurodegenerative disorders such as Alzheimer’s disease, etc. [8,9]. Designing treatments for some of these diseases should necessarily involve therapies to preserve/regenerate myelin. To this end, it appears crucial to understand the mechanisms governing myelin generation, maintenance, and regeneration.

Oligodendrocytes are generated by differentiation of oligodendrocyte progenitor cells (OPCs) that themselves originate from defined germinal niches in the embryonic brain and spinal cord [10]. OPCs colonize CNS parenchyma in successive waves [11]. A proportion of these cells myelinates the axons, while another subset remains undifferentiated and constitutes the population of adult OPCs (aOPCs or so-called NG2 cells) [12]. Adult OPCs represent 4–5% of the total CNS cells. In general, aOPCs proliferate at a low rate, and differentiate and contribute to myelin turnover throughout adult life [13,14]. These processes underlie the functional plasticity of the CNS in response to motor learning, for example [14].

One of the most studied functions of aOPCs is myelin regeneration following demyelinating insults. Using animal models of demyelination, it has been shown that, upon myelin loss, aOPCs undergo transcriptional changes that induce transition into so-called “activated” state, and their proliferation rate increases. These cells migrate towards demyelinated axons in response to defined cues, where they differentiate into oligodendrocytes and regenerate myelin [12]. Interestingly, it has been shown that the process of (re)myelination involves synaptic communication between axons and OPCs [15,16,17].

Therefore, a number of events are required for an OPC to successfully perform CNS (re)myelination, namely: transcriptional activation of defined genes, proliferation in most cases, migration towards axons to be (re)myelinated, synaptic communication with these axons, differentiation into oligodendrocytes, and lipid and protein synthesis for myelin (re)generation. These processes require energy (in the form of ATP) and building blocks for lipids and proteins, which implies a tremendous investment of energy fuels and molecular sources of carbon. In terms of energy, it has been estimated that OPCs generates three times its own weight in membrane per day during myelination [18], which is likely to be similar during myelin regeneration.

Once myelin is generated by an oligodendrocyte, it has to be maintained. It has been estimated that oligodendrocytes maintain up to 100 times their own weight in membrane throughout life [18]. Given that myelin maintenance is a dynamic process that involves continuous lipid [19,20] and protein synthesis [21], oligodendrocytes are likely to require a significant energetic/building block input throughout life. In addition, in the last decade, it has been shown that oligodendrocytes provide axons with energy fuels [2], which adds an additional energetic cost to the proper functioning of these cells.

Deciphering energetic demands and the sources of energy/building blocks used by oligodendroglial cells (OPCs and mature oligodendrocytes) is, therefore, of major importance for understanding myelin pathology in various CNS diseases and designing treatments. Importantly, bioenergetic failure has been suggested as a cause of neurodegeneration in several CNS diseases, including Parkinson’s disease, Alzheimer’s disease, and Huntington’s disease [22,23,24,25]. Regarding MS, several studies have highlighted impairments in energy metabolism in active and inactive lesions, as well as in the normal-appearing white matter in several cell types, including oligodendroglia [26,27,28,29,30,31,32,33,34,35,36]. Moreover, depletion of ATP metabolites in the cerebrospinal fluid of MS patients is correlated with a more severe disability progression [28]. Thus, it has been hypothesized that oligodendroglial pathology/dysfunction and the development of permanent disability in MS result, at least in part, from energy failure.

This review summarizes our current knowledge of energy metabolism pathways in oligodendroglial cells, how these may differ during the process of differentiation, and which energy sources might be crucial to sustain (re)myelination.

## 2. Oligodendroglial Functions Require Energy and Biosynthetic Precursors

The primary functions of oligodendroglia are myelination and trophic support to the axons.

### 2.1. Energetic Cost of Myelin Synthesis

Myelination consists of biosynthetic reactions, namely lipid and protein synthesis, which require input of simpler precursor molecules and consume energy.

#### 2.1.1. Lipid Synthesis

As mentioned above, lipids are the major component of myelin membranes (70–80% of dry weight). The composition of myelin is unique as compared to other membranes, with an approximate 2:2:1 ratio for cholesterol:phospholipids:glycolipids [37]. Myelin contains 80% of the cholesterol present in the brain. Blocking oligodendroglial cholesterol synthesis in mice significantly delays myelination, even though this process eventually achieves completion, thus suggesting that oligodendrocytes can obtain cholesterol by compensatory mechanisms [38,39]. Indeed, it has been shown that myelination can be sustained in part by astrocyte-derived lipids [40]. Unlike cholesterol, other lipids are built using fatty acids as building blocks [20]. Blocking endogenous oligodendroglial fatty acid synthesis impairs myelination and remyelination, which can be only partially rescued by dietary fatty acid supplementation [41], thus highlighting the importance of oligodendroglial fatty acid synthesis in myelination. Moreover, continuous lipid synthesis in oligodendrocytes is required for myelin maintenance [19].

The first step of fatty acid synthesis is the synthesis of palmitate from acetyl CoA. This process is fundamental in myelin generation, as the knockout of oligodendroglial fatty acid synthase, the enzyme complex that carries out this process, impairs myelin generation and repair [41]. The synthesis of one molecule of palmitate consumes 8 molecules of acetyl CoA, 7 ATPs, and 14 NADPH, as well as 6 protons. In addition, as estimated by Harris and Attwell [42], if one takes into account the loss of ATP generation when acetyl CoA molecules are diverted from oxidative phosphorylation (ATP generation) to palmitate synthesis and glucose molecules are diverted to PPP from glycolysis (to generate NADPH required), the synthesis of palmitate costs 228 ATP molecules. The synthesis of one molecule of phosphatidylethanolamine (a phospholipid) would cost 542 ATPs (equivalent to oxidative phosphorylation of 18 glucose molecules) and galactosyl cerebroside (a galactolipid, a subtype of glycolipids) 535 ATP molecules. Given that cholesterol used in myelin cannot be imported from the circulation as it cannot cross the blood–brain barrier [43], it has to be synthesized. The energetic cost for its synthesis would be of 306 ATPs per molecule. Taking into account the normal lipid composition of myelin, the lipid production required for one gram of myelin was estimated to cost 3.24 × 10^23^ ATP molecules [42].

#### 2.1.2. Protein Synthesis

Myelin contains a defined set of proteins, the most abundant ones in the CNS being myelin basic protein (MBP) and the proteolipid protein (PLP/DM20). The mechanism underlying the synthesis of PLP is quite different from that of MBP synthesis. PLP is synthesized at the rough endoplasmic reticulum, processed through the Golgi apparatus, and transported via vesicles to the myelin membrane. However, in the case of MBP, mRNA is transported along oligodendrocytes processes, and then translated locally at the axon-oligodendrocyte contact site [44,45,46]. Thus, the energetic cost of the synthesis of CNS myelin proteins includes the energy invested in peptide chain elongation, in addition to the cost of mRNA transport along oligodendrocyte processes and local translation (in the case of MBP), or that of vesicular transport (PLP).

Assuming that aminoacids were used only for protein synthesis and not as intermediates in the Kreb’s cycle, and that proteins represent 25% of myelin, Harris and Attwell [42] calculated that the cost of protein production per gram of myelin should be of 6.84 × 10^21^ ATP molecules. This amount does not include the cost of protein targeting to the right place within the myelin sheath.

Thus, in order to carry out lipid and protein synthesis, myelinating cells will require a significant input of ATP, as described above, as well as extensive amounts of carbon skeletons as building blocks.

### 2.2. Oligodendroglial Trophic Support to the Axons Requieres Energy Investment and/or Metabolite Shuttling

The notion that myelin may have functions other than that of insulator (for acceleration of axonal conduction) arouse from the observations that mouse mutants for certain myelin proteins show deficits in axonal transport and axonal degeneration in the absence of demyelination [4,5]. This led to hypotheses that oligodendrocytes provide axons with trophic support. Such possibility would explain how axons sustain their energy metabolism given that they are spatially separated from the extracellular space by the myelin sheath, except at the nodes of Ranvier.

Several modes of oligodendroglial trophic support to the axons have been described. The first one, which has received much attention, is the so-called “oligodendrocyte-axon lactate shuttle,” by which oligodendrocytes metabolize glucose to pyruvate through glycolysis, convert pyruvate to lactate, and then export lactate via monocarboxylate transporter 1 (MCT1), expressed on myelin [47,48], to the axons [47] (Figure 1). An alternative explanation for this lactate shuttling could be that oligodendrocytes may serve as a tunnel, by importing lactate (through MCT1 that can both import and export lactate, or through gap junctions from astrocytes) and shuttling it to the axons, rather than performing extensive rounds of glycolysis (Figure 1).

The second mode of trophic support has been described by Meyer and colleagues, who found that oligodendrocytes in the corpus callosum sustain compound action potentials by delivering both glucose and lactate to the axons, and that pan-glial networks via connexin 47 are important in this process [49]. Moreover, thalamic oligodendrocytes have recently been shown to assist astrocytes in delivering glucose and lactate to the axons to support their synaptic activity [50]. The transport of metabolites from oligodendrocyte soma to the axons likely involves networks of interconnected cytoplasmic pockets observed in the myelin using high-pressure freezing electron microscopy [7].

Axonal trophic support by oligodendrocytes may also be carried out via myelin peroxisomes, potentially by fatty acid transfer for beta oxidation in the axons [51]. Recently, it has also been shown that oligodendrocytes support axonal transport and maintain the health of nutrient-deprived axons in vitro via exosomes [52].

While the exact nature of oligodendroglial trophic support to the axons still remains to be clarified, it is clear that the ability to provide nutrients to the axons will impose additional energetic demands on oligodendroglia. The notion that oligodendrocytes supply axons with lactate implies that these cells need to either produce lactate through glycolysis or import it from the extracellular space and shuttle it to the axons. In order to provide the axons with glucose, as shown by two reports [49,50], oligodendrocytes would have to either import glucose (from the extracellular space or from astrocytes via gap junctions) (Figure 1) and shuttle it to the axons, or produce glucose by gluconeogenesis. Yet, there is no evidence that gluconeogenesis occurs in oligodendrocytes.

## 3. Energy Fuels Used by Oligodendroglia

As mentioned above, myelin synthesis and trophic support to the axons by mature oligodendrocytes imply an extensive need for building blocks and ATP. Regarding differentiating OPCs, energy fuels are required to sustain migration to demyelinated lesions, proliferation, and myelin synthesis. While glucose is the main source of energy for all cells, alternative fuels may also be important to sustain the energetic needs of oligodendroglia.

### 3.1. Glucose

#### 3.1.1. Metabolizing Pathways

Glucose can be metabolized in the glycolytic pathway (which in the presence of oxygen is in most cases followed by oxidative phosphorylation), pentose phosphate pathway, or hexosamine pathway (Figure 2).

**Glycolysis** is the conversion of one molecule of glucose into two molecules of pyruvate, two hydrogen ions, and two molecules of water. This process consumes two ATP molecules and generates four, resulting in the net gain of two ATP molecules per one molecule of glucose, which means that it is not very efficient on its own in producing energy. It also leads to production of two NADH molecules. Glycolysis is not oxygen-dependent.

**Oxidative phosphorylation**: In the presence of oxygen, the pyruvate produced by glycolysis is converted to acetyl Co-A, which is then oxidized in the Kreb’s cycle (tricarboxylic acid (TCA) cycle), to produce CO2, in addition to NADH and FADH2, the electron carriers that pass the electrons to the electron transport chain. The reactions occurring in the electron transport chain are denominated oxidative phosphorylation. This pathway is the most efficient ATP-producing pathway (24–28 molecules per glucose molecule).

In the absence of oxygen, cells continuously undergo glycolysis in order to produce ATP, and pyruvate is converted to lactate (or to ethanol in yeast). Lactate is then exported from the cell. However, in some cases (for example, in cancer cells), cells preferentially undergo glycolysis and export lactate even in the presence of oxygen. This phenomenon is denominated the Warburg effect.

**The pentose phosphate pathway (PPP)** is a glucose-metabolizing biosynthetic pathway that neither consumes nor produces ATP, and it does not require oxygen. The first phase of this pathway is oxidative, and it converts NADP to NADPH, which will be used for (a) biosynthetic reactions as a reducing agent (e.g., fatty acid and sterol synthesis) and (b) to prevent oxidative stress (production of glutathione and thioredoxin). The second phase of PPP is a non-oxidative synthesis of 5-carbon sugar (riboses), used in the synthesis of nucleotides and nucleic acids. Depending on the cellular needs, for example, if it needs much more NADPH (fatty acid synthesis) than nucleotide production, after step 1 (oxidative phase), the intermediates produced can enter the glycolytic pathway, resulting in the production of pyruvate rather than continuing in the second phase.

**The hexosamine pathway** is a biosynthetic pathway that branches out of glycolysis and converts fructose-6-phosphate into a key substrate for protein glycosylation, uridine diphosphate N-acetyl glucosamine (UDP-GlcNAc). This pathway is crucial for the biosynthesis of complex molecules such as glycoproteins, glycolipids, proteoglycans, and glycosaminoglycans. It consumes approximately 2–5% of glucose that enters the cell. It neither consumes nor produces ATP.

#### 3.1.2. Glucose Metabolism in Oligodendroglia: Variations across the Lineage

Glucose enters the cell through glucose transporters. In the brain, the expression of facilitative diffusion glucose transporters GLUT1–6 and GLUT8 and the Na+-D-glucose cotransporter SGLT1 has been demonstrated [53]. While gene expression data suggest that OPCs may express Glut3, a high affinity transporter also expressed in neurons [54], protein expression pattern of Glut expression on OPCs remains to be investigated. Oligodendrocytes have been reported to express Glut1 [55], a transporter with high affinity, although somewhat lower than that of Glut3 [56,57] and potentially also Glut2 [58,59]. In addition, the expression of Glut1 by mature oligodendrocytes in vitro was increased following stimulation with glutamate NMDA receptor agonists [55], therefore highlighting that axonal electrical activity, a fundamental inducer of myelination [60], increases metabolic activity of oligodendrocytes. Importantly, Glut1 expression in rat oligodendrocytes was reduced upon incubation of these cells with antibodies or cerebrospinal fluid (CSF) isolated from the patients suffering from anti-NMDA receptor encephalitis, suggesting reduced oligodendroglial metabolic activity in the disease [61].

#### 3.1.3. Glycolysis versus Mitochondrial Metabolism

Numerous studies have examined the main pathways elicited by glucose in oligodendroglia, and the results reported are somewhat contradictory, mainly regarding the extent of mitochondrial metabolism in mature oligodendrocytes. The experiments on cells isolated from the developing brain showed that oligodendrocytes oxidize twice as much glucose as astrocytes in the Kreb’s cycle, just slightly less than neurons [62,63], thus implying high rates of mitochondrial metabolism. In contrast, an in vivo study [64] showed that deletion of Cox10 in post-mitotic oligodendrocytes, which prevents these cells from forming a stable mitochondrial complex IV, did not affect CNS myelination nor axonal integrity, thus suggesting that mitochondrial activity was not required to maintain mature oligodendrocyte viability nor myelination in vivo. It is interesting to note that studies of neuronal responses to mitochondrial mutations in vivo showed that Cox10 mutants had prolonged survival (10–12 months) and less reactive oxygen species damage as compared to the mitochondrial complex III mutants [65]. This suggests that Cox10 deletion, at least in neurons, may not have completely abolished mitochondrial activity. Moreover, it has been argued that mitochondrial defect in oligodendrocytes may become compensated by intact astrocytic mitochondria [66]. Thus, further studies should be performed to confirm whether mitochondrial metabolism in myelinating oligodendrocytes indeed becomes dispensable in vivo.

Interestingly, in vitro studies have shown that the rate of glycolysis versus oxidative phosphorylation/mitochondrial metabolism depends on the differentiation stage of oligodendroglial cells. Thus, inhibition of mitochondrial complex IV lead to more extensive injury of MBP-expressing oligodendrocytes than OPCs (both derived from post-natal rat brain) [67], suggesting more extensive complex IV activity in mature cells in vitro. The treatment of human cells with rotenone (complex I inhibitor) did not affect the survival of undifferentiated OPCs nor differentiated oligodendrocytes, but it compromised the viability of differentiating oligodendroglia [68], suggesting that newly generated oligodendrocytes/late OPCs are the most vulnerable to complex I inhibition. In a study using Seahorse analyzer to evaluate glycolytic versus mitochondrial ATP production, Rao and colleagues observed that, under optimal conditions, cultured OPCs and oligodendrocytes derived from postnatal rat brain rely on oxidative phosphorylation for ATP synthesis, while oligodendrocytes isolated from the adult rat brain preferentially use glycolysis [69], which appears consistent with the lack of myelinating oligodendrocyte pathology following Cox10 deletion in vivo [64]. Following glucose deprivation, adult rat oligodendrocytes reduced glycolysis and increased oxidative phosphorylation [69]. Using the same technique (Seahorse analyzer), Antel group also found that both human OPCs and oligodendrocytes isolated from adult tissue use glycolysis to generate ATP under optimal conditions in vitro [70]. Moreover, both rat and human OPCs had higher oxygen consumption compared to oligodendrocytes, and human cells (OPCs and oligodendrocytes) showed less ATP production than cells derived from post-natal rat brain [70]. When exposed to nutrient deprivation, human oligodendrocytes decreased ATP production, which was not the case with OPCs. Therefore, it appears that, at least in vitro, both rat and human oligodendroglia show differences in energy metabolism and metabolic adaptations to nutrient stress according to the differentiation state and the age of the tissue the cells were derived from.

#### 3.1.4. Pentose Phosphate Pathway

As mentioned above, glucose can also enter the PPP pathway. Amaral and colleagues showed, using C13 tracing studies, that oligodendrocytes metabolize 10–15% of the glucose in the PPP [63], which is comparable to astrocytic PPP activity. This rate of the PPP pathway is lower than the one in cell cultures with increased proportion of undifferentiated OPCs [71], reported to be two-fold higher than that in astrocytes and four-fold higher than in neurons (reviewed by Amaral and colleagues [72]). This would be consistent with the fact that proliferation in cultures with less differentiated cells is higher, and therefore, requires more nucleotides (the generation of which requires PPP). However, as indicated by Amaral and colleagues in a comprehensive and highly informative review of glucose metabolism in oligodendroglial cells [72], it is possible that PPP activity in mature oligodendrocytes may be higher than what was estimated, because the contribution of glucose metabolized in the PPP to the lipid metabolism was not examined. Importantly, it has been reported that de novo fatty acid and cholesterol synthesis are closely related to the rate of PPP in oligodendroglial cells in the developing brain [73].

#### 3.1.5. Hexosamine Pathway

The hexosamine pathway is likely to play an important role in myelination. As mentioned above, this biosynthetic pathway leads to the production of glycoproteins, glycolipids, proteoglycans, and glycosaminoglycans. Importantly, several protein components of myelin are glycoproteins (such as myelin associated glycoprotein (MAG) and myelin oligodendrocyte glycoprotein (MOG)), and glycolipids are a fundamental component of myelin. Moreover, proteoglycans such as NG2 are important determinants of OPC migration [74]. Indeed, it has been reported that N-glycan branching of GlcNAc is important in myelination and myelin repair [75].

## 4. Alternative Sources of Energy and Biosynthetic Precursors

As mentioned above, glycolysis, PPP, and hexosamine pathways are strictly fueled by glucose. However, the pathways using acetyl CoA, such as Kreb’s cycle/oxidative phosphorylation and lipid synthesis, can also be directly fueled by alternative sources (Figure 2). These include monocarboxylates and N-acetyl-aspartate (NAA), and appear particularly important during developmental myelination (Figure 3).

### 4.1. Monocarboxylates (Ketone Bodies and Lactate) and Their Transporters in Oligodendroglia

The most important alternative fuels for the brain appear to be monocarboxylates, a family of molecules that includes pyruvate, lactate, and ketone bodies (acetoacetate and β-hydroxybutyrate). It is known that some of these substrates are essential for sustaining brain function in specific situations, so that ketone bodies are used as an alternative energy source during starvation or lactation while lactate represents an ATP source during intense physical activity and in the newborn brain early after delivery [76]. In the past few decades, lactate has also emerged as an important determinant of constitutive neuronal function as it has been described as a preferred energy substrate (over glucose) for neuronal oxidative metabolism in vitro [77], and more recently, in vivo [78,79]. Even though the brain can take up lactate from systemic circulation under specific circumstances described above, constitutive neuronal function is mostly sustained by lactate originating from two glycolytic intra-cerebral cellular sources. Thus, astrocytes, in response to synaptic activity, increase glycolysis, convert pyruvate to lactate, and shuttle lactate to neurons [80]. As mentioned above, oligodendrocytes have also been shown to supply lactate to myelinated axons, which appears crucial for neuronal activity and survival [50,81].

Monocarboxylate trafficking into/out of the cell requires monocarboxylate transporters (MCTs) [82]. MCTs 1, 2, and 4 are expressed in the CNS, and they co-transport monocarboxylates with a proton. MCT1 is the intermediate affinity transporter, which means that it both imports and exports monocarboxylates. MCT2 is the highest affinity transporter, and functions to import monocarboxylates into the cell. MCT4 is a high affinity transporter, but it always exports lactate, even in high-lactate environments [76,83]). The initial studies of MCT expression in the CNS demonstrated MCT1 as the most ubiquitously expressed transporter, while MCT4 expression was described predominantly on astrocytes, and that of MCT2 on neurons [82]. Regarding oligodendroglia, MCT1 expression has been detected on myelin in vivo [48], which is important for lactate shuttling to the axons [47].

The fact that oligodendrocytes express MCT1 suggests they can both import and export monocarboxylates. So far, this transporter has been mostly implicated in lactate export to myelinated axons. Its expression was reported on oligodendrocytes but not OPCs in one paper [47], but in another report, OPC upregulation of MCT1 was observed following mild oxygen-glucose deprivation [84]. Viral-mediated deletion of MCT1 in myelinating oligodendrocytes lead to axonal degeneration in absence of demyelination, suggesting its predominant role in oligodendroglial trophic support to the axons [47]. Moreover, decreased expression of this transporter in oligodendrocytes has been reported in human brains during neurodegenerative diseases such as Creutzefeld Jacob [85], Alzheimer’s disease [86,87], and Amyotrophic Lateral Sclerosis (ALS) [87]. A recent paper examined the effects of conditional MCT1 deletion in oligodendroglial lineage and myelinating oligodendrocytes [88]. The results show that loss of MCT1 did not affect developmental myelination or early axonal energy homeostasis. Evidence of axonal degeneration and hypomyelination was observed quite late, at 360 days of age, and was accompanied by an increase in microglial reactivity. Only at 750 days of age did axonal degeneration become prominent. Thus, the authors concluded that monocarboxylate metabolism is not an important determinant of oligodendroglial energy metabolism, but that it becomes important with age, when loss of MCT1 transporter on oligodendrocytes enhances age-induced degeneration. It is important to mention that monocarboxylates (and other metabolites) could in theory also be transferred via connexin channels from astrocytes to oligodendroglia [49,89,90] (Figure 1) or through other MCTs. Importantly, connexin 30 (astrocytic form) and connexin 47 (oligodendrocyte form) double mutants exhibit hypomyelination [91]. Moreover, the importance of pan-glial networks via connexin channels in oligodendroglial support of axonal metabolism has been shown [49,50]. Thus, the absence of MCT1 in oligodendroglia may have been compensated by monocarboxylate transport through connexin channels. Moreover, in the recent years, studies of gene expression have suggested that the cells of oligodendrocyte lineage may also express other MCTs [54], suggesting that these could also participate in the oligodendroglial monocarboxylate metabolism.

#### 4.1.1. Ketone Bodies as an Important Energy Fuel during Developmental Myelination

Experiments performed in 1970s–1980s showed that, in the developing brain, ketone bodies appear to be an important substrate for myelin synthesis (Figure 3). Page, Krebs and colleagues demonstrated that the blood levels of ketone bodies in the developing rats exceeded 6–10 times those in adult animals, and that levels of enzymes involved in ketone body utilization were three times higher in rats at weaning than in adult rats [92,93]. Importantly, ketone bodies were at least as important as glucose in terms of energy fuels [92]. This observation appears consistent with the fact that the developing brain may account for up to 60% of the basal energetic demands of the body [94] as compared to 20% used by the adult brain [95], which suggests increased energetic needs that may require additional fuels to complement energetic inputs by glucose. Interestingly, administration of radioactively labeled glucose versus ketone bodies to young rats (18–21 days old) showed that C14 incorporation into sterols and fatty acids in myelin (and in subcellular fractions) occurred to a higher extent following ketone body as compared to glucose administration, thus suggesting that ketone bodies are the preferential source of acetyl coA used for myelin lipid synthesis [96]. While in this report, acetoacetate and D-3-hydroxybutyrate were used to a similar extent during lipid synthesis, experiments performed on newborn rats showed a preferential incorporation of β-hydroxybutyrate [97].

**Figure 3 life-11-00238-f003:**
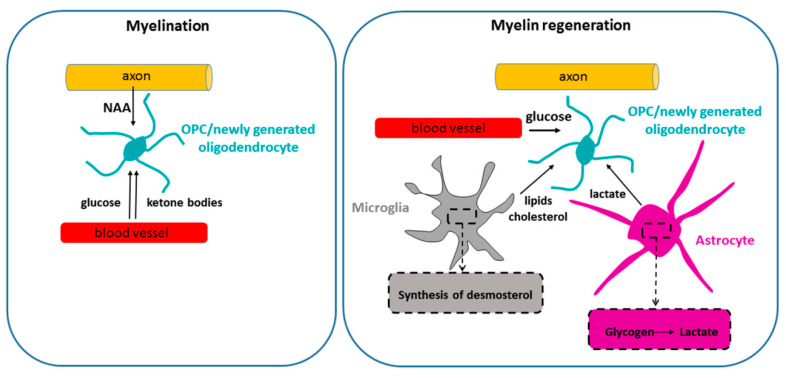
Metabolic support during postnatal myelination and remyelination. During postnatal development, myelination is fueled by glucose and at least two other energy sources. Thus, ketone bodies, imported from the circulation, are used as myelin lipid precursors by neonatal OPCs/oligodendrocytes, and potentially also for ATP generation. Neuron-derived N acetyl aspartate (NAA), released by the axons and taken up by neonatal oligodendroglial cells, also plays an important role in lipid synthesis during developmental myelination. It is unknown whether the availability of NAA and ketone bodies plays a role in remyelination. Remyelinating adult OPCs/new oligodendrocytes have been reported to rely on glycogen-derived lactate released by astrocytes in a cuprizone model [98]. In addition, synthesis of desmosterol by microglia has been shown to stimulate lipid recycling after demyelination and drive remyelination [99].

Experiments in cell culture also showed incorporation of ketone bodies into lipids by oligodendrocytes [100]. In addition, oligodendrocytes (as well as neurons and astrocytes) isolated from the developing brain used ketone bodies to a 7–9 times greater extent than glucose for oxidative metabolism, thus indicating that these cells may spare glucose for pathways that cannot be fulfilled by fat metabolites [62] (Figure 2). A separate study also reported that oligodendroglial oxidative metabolism and fatty acid and cholesterol synthesis rely predominantly on ketone bodies and pyruvate, while glucose is rapidly converted to lactate [101].

Levels of ketone bodies in the blood can be raised by administration of a ketogenic (low-carbohydrate) diet [102]. Beneficial effects of a ketogenic diet on myelination have been shown in a mouse model of Pelizaeus-Merzbacher disease (a fatal and incurable leukodystrophy) with preserved blood–brain barrier [103]. In this model, a ketogenic diet restored oligodendrocyte integrity and myelination, and reduced axonal degeneration. Moreover, benefits of ketogenic diet were also reported in a patient with AGC1 deficiency, the disease characterized by secondary hypomyelination due to a lack of N-acetylaspartate (NAA) and defective oligodendroglial fatty acid metabolism. Administration of a ketogenic diet to this patient led to a remarkable increase in myelination [104]. A ketogenic diet can influence not only oligodendrocyte integrity, myelination, and axonal survival, but also the inflammatory cell profile in the blood [105,106]. Slight improvements were observed in clinical trials testing the effect of ketogenic diet in patients with MS, including a reduction in inflammatory markers and a slight reduction in clinical scores in some cases [106].

#### 4.1.2. Lactate Metabolism in Oligodendroglia: Differentiation State Matters?

Lactate is another potential alternative energy fuel for oligodendrocyte lineage cells. Rinholm and colleagues reported that lactate rescues myelination in cerebellar slices when these are exposed to glucose deprivation [48]. It is not completely clear whether in these experiments lactate stimulated oligodendrocyte differentiation and myelination directly, by providing oligodendroglial cells with energy or carbon skeletons for myelin synthesis, or indirectly, by sustaining axonal activity which stimulated myelination. However, by pH imaging (lactate transport is coupled to that of a proton), the authors observed that the main lactate importers were oligodendrocytes, suggesting that the stimulation of myelination was at least in part due to oligodendroglial import of lactate. Abarca and colleagues also showed that lactate was utilized by oligodendrocytes in vitro as energy fuel and a lipid precursor [71]. These authors administered C14-labeled lactate to oligodendroglial, astrocytic, and neuronal cell cultures and observed a six-fold higher incorporation of C14 into lipids by oligodendrocytes as compared to neurons and type-2 astrocytes. Ichihara and colleagues showed that lactate was able to rescue OPC cycling and differentiation under low glucose conditions, which was inhibited by a lactate transporter inhibitor, α-cyano-4-hydroxy-cinnamate [98]. The identity of the lactate transporter responsible for lactate import into oligodendroglia was not investigated in these experiments. On the other hand, Lee and colleagues showed that in vivo, oligodendrocytes export lactate to the axons as a trophic support [47]. The apparently contradictory findings between the reports in vitro/ex vivo claiming lactate utilization by oligodendroglia [48,71] and those in vivo showing lactate export by glycolytic oligodendrocytes [47,64] could be interpreted as a change in the metabolism of oligodendrocytes induced by myelination. However, it could also be that the availability of lactate allows oligodendrocytes to sustain oxidative phosphorylation and lipid synthesis, and thus, use glucose preferentially for lactate production (and delivery to the axons), pentose phosphate pathway, and hexosamine pathway (Figure 2). Accordingly, the paper by Amaral and colleagues showed that differentiated oligodendrocytes incorporate C13-labeled lactate, and that the presence of this alternative fuel stimulates glucose utilization via glycolysis, and consequently, lactate production [63]. Such compartmentalization of energy metabolism that spares glucose for functions that can solely be fulfilled by this sugar (Figure 2) has also been suggested in the case of ketone bodies [101]. Thus, it is possible that myelinating oligodendrocytes both use lactate for lipid synthesis and produce lactate through glycolysis (Figure 1 and Figure 2).

It should be kept in mind that lactate can function not only as a metabolite, but also as a signaling molecule [107]. Lactate receptor, a G-Protein-Coupled Receptor 81/Hydroxycarboxylic Acid Receptor 1, is expressed in the brain [108]. While no reports so far have described GPR81/HDAC1 expression specifically on oligodendroglia, the possibility that lactate may affect oligodendroglial physiology as a signaling molecule rather than as a metabolite cannot be excluded. It is, therefore, important to distinguish, as much as possible, between the role of lactate as a signaling molecule versus that of an energy fuel, which is possible experimentally by comparing the effect of L-lactate versus D-lactate (a non-metabolizing molecule), and by performing experiments using lactate transporter inhibitors, which inhibit lactate import/export but not signaling.

### 4.2. N-Acetyl-Aspartate Sustains Developmental Myelination

Acetyl CoA used by oligodendrocytes for lipid synthesis and/or ATP production can be derived from glucose and monocarboxylates, but also from N-acetylaspartate (NAA) (Figure 2). NAA is produced by neurons, and aspartoacetylase (ASPA), the enzyme that catabolyzes NAA into L-aspartate and acetate, is expressed by oligodendrocytes [109]. The transfer of NAA from the axons to oligodendrocytes represents an important metabolic support for oligodendrocytes during myelination [110,111]. Loss of ASPA in oligodendrocytes renders these cells incapable of converting NAA to acetate, which leads to failure of lipid synthesis and myelination observed in a leukodystrophy called Canavan’s disease [112]. Studies in ASPA-deficient tremor rat showed that deficient NAA metabolism leads predominantly to myelin lipid abnormalities, indicating that NAA is an important substrate for myelin lipid synthesis in the developing brain [113]. Moreover, studies of ASPA-deficient mice show a decrease in myelin-associated lipids during postnatal myelination [112]. An in vitro study also suggested that aspartate derived from NAA is used predominantly for lipid synthesis [114]. Yet, studies in a mouse model of Canavan´s disease showed that ASPA loss affects both lipid synthesis and ATP production, thus suggesting that NAA represents both a lipid precursor and fuel for ATP synthesis, which appears particularly important during developmental myelination [115]. Lastly, ASPA deletion specifically in oligodendrocyte lineage in vivo resulted into hypomyelination, and decreased galactosylceramide amounts in brain homogenates [116].

Thus, axons provide a significant metabolic support for oligodendrocytes via NAA during developmental myelination (Figure 3). It remains to be determined whether this mode of metabolic communication plays a role in myelin maintenance and/or remyelination.

## 5. Energetic Support of Remyelination?

Remyelination is a process by which lost myelin sheaths are regenerated around the axons that have undergone demyelination. This process is highly efficient in young laboratory animals, but its rate slows down with age [117,118]. Remyelination is neuroprotective in animal models and patients suffering from MS [3,119,120], which is why designing strategies to stimulate this process appears crucial to prevent/diminish the development of permanent neurological handicap associated with chronic MS.

Very little is known about metabolic requirements of CNS remyelination (Figure 3). A recent paper suggested that microglial synthesis of desmosterol, a cholesterol precursor, was required to support remyelination in mice subjected to cuprizone intoxication by stimulating lipid recycling for myelin synthesis [99]. This paper also suggested that OPC synthesis of cholesterol was not crucial for remyelination. Another paper suggested that glycogen-derived lactate plays a role in remyelination in a cuprizone model [98]. However, it was not clear whether this effect was mediated by lactate support of neuronal activity (that stimulates myelination) or direct support to OPCs/differentiating oligodendrocytes. Thus, deciphering the energetic mechanisms underlying the process of CNS remyelination is a major challenge.

Importantly, the notion that the sole effectors of successful remyelination are OPCs has been challenged. Thus, CNS demyelination in cats fed with irradiated diet and vitamin B12-deficient monkeys occurred because oligodendrocytes stopped myelinating, even though they did not die [121]. Importantly, remyelination in these models was carried out by surviving mature oligodendrocytes [122]. These results have important implications for the subtype of demyelinated MS lesions in which mature oligodendrocytes are preserved [123], as they indicate that oligodendrocytes that survive demyelination can participate in remyelination. It remains to be determined whether this type of demyelination may be a consequence of dysregulated energy homeostasis in the CNS. Notably, it has been shown that acute MS lesions are hypermetabolic [124], which suggests that ongoing inflammatory activity may consume local energy sources, and thus, potentially compromise the energetic support for (re)myelinating cells.

## 6. Distinct Differentiation Stage-Specific Responses of Oligodendroglia to Nutrient Deprivation: Relevance to Demyelinating Pathology

Several studies have examined the responses of oligodendroglial cells to metabolic stress. Exposure of rodent oligodendroglia to glucose deprivation induced extensive changes in the processes of OPCs but not in those of mature oligodendrocytes [84]. Another study showed that under low glucose conditions in vitro OPC survival, proliferation, and differentiation were decreased but could be rescued by lactate supplementation [98]. Studies using human oligodendroglia showed that exposure of OPCs and oligodendrocytes to low glucose conditions initially leads to marginal cell death accompanied by extensive process retraction by oligodendrocytes, which was associated with a decrease in the rate of glycolytic metabolism [70]. Significant apoptosis in this report was observed only after 6 days of glucose deprivation. A more recent study showed that the age of the tissue the human cells were derived from determines the susceptibility of oligodendroglial cells to nutrient-induced apoptosis. Thus, fetal-derived human oligodendroglia were the most susceptible to apoptosis after nutrient deprivation, and pediatric cells were more susceptible than adult-derived oligodendrocytes [125]. The survival of adult human tissue-derived oligodendrocytes upon nutrient deprivation was related to an upregulation of anti-apoptotic Bcl-2 family. Moreover, under these conditions, adult human oligodendrocytes upregulated autophagy (a process that can mobilize cellular energy stores [126]), which maintained their survival in the short term. Long-term exposure of adult human oligodendrocytes to nutrient deprivation led to cell death that was different from apoptosis [125]. Importantly, the main response of adult oligodendrocytes to nutrient deprivation was process retraction and low metabolic rate, which compromised myelination, but preserved survival [70]. This in vitro phenomenon has been associated with distal process retraction by oligodendrocytes, previously described as dying-back oligodendropathy [127], and demyelination observed in a subtype of MS lesions [121]. This pathology appears reversible until a specific point of no return is reached [121], suggesting that oligodendrocytes that stop myelinating in response to metabolic stress may recover their myelinating function if optimal conditions are restored. Interestingly, exposure of cats to irradiated diet led to demyelination, and the return to normal diet was followed by remyelination by pre-existing oligodendrocytes [122,128]. Thus, it may be that the initial response of myelinating oligodendrocytes to metabolic stress is to retract processes and stop myelinating in order to save the energy for survival. Such lack of myelination may be reversible if nutrient deprivation does not extend in time. However, long-term deprivation leads to mature oligodendrocyte death that is different from apoptosis (Figure 4). Interestingly, a study of aging in the female brain reported that white matter lipids may be used as a source of energy [129], thus suggesting that energy-deprived oligodendrocytes might even oxidize myelin fat to obtain energy for survival. Such observations have important implications for strategies to prevent myelin loss/stimulate myelin repair in multiple sclerosis, and should be addressed by future studies.

## 7. Conclusions

Oligodendrocytes and their progenitors require a significant input of carbon sources and energy fuels to sustain myelin synthesis and ATP production. Moreover, myelinating oligodendrocytes also invest energy to provide trophic support to the axons. It is likely that, in addition to glucose, complementary energy fuels might be required to fulfill oligodendroglial energetic needs and assure correct myelination and axonal support. Unravelling the mechanisms underlying oligodendroglial energy metabolism should increase our understanding of oligodendroglial pathology in demyelinating diseases, and thus, provide novel clues for therapies that aim to prevent myelin loss/enhance myelin regeneration.

## Figures and Tables

**Figure 1 life-11-00238-f001:**
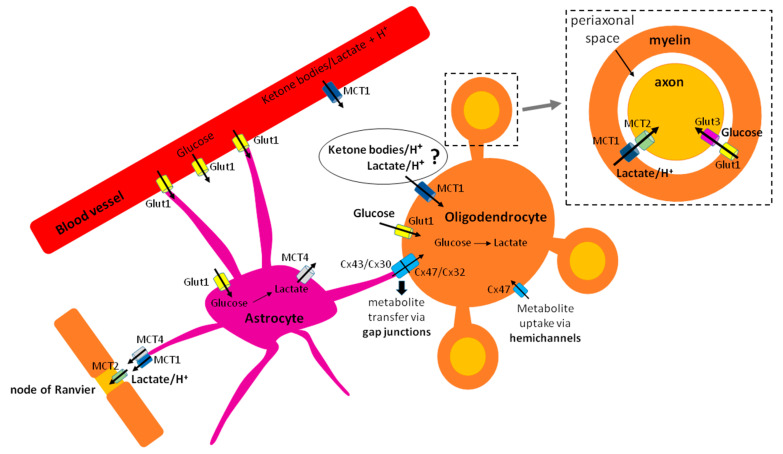
Metabolite shuttling between astrocytes, myelinating oligodendrocytes, and axons optimizes energy fuel use in the central nervous system white matter. Glucose is imported into the parenchyma from the blood through Glut1 transporter on endothelial cells, or directly into astrocytes through the same transporter. Astrocytes convert glucose to lactate and export lactate to the extracellular space through MCT4. In addition, lactate as well as ketone bodies can also be imported from the circulation into the parenchyma through MCT1 on endothelial cells. Oligodendrocytes can import glucose from the extracellular space via Glut1 and lactate/ketone bodies via MCT1. Metabolites could also potentially enter oligodendrocytes via connexin hemichannels. In addition, astrocytic lactate and glucose can be transferred to oligodendrocytes via gap junctions formed by dimers of astrocytic connexins 43 and 30 with oligodendrocyte connexins 47 and 32, respectively. In order to provide axons with energy fuels, oligodendrocytes metabolize glucose to pyruvate through glycolysis, then convert pyruvate to lactate, and export lactate to the periaxonal space via MCT1 expressed on myelin. Axons then import lactate via MCT2. Oligodendrocytes may also export glucose to the periaxonal space via Glut1 that can then be uptaken by the axon via Glut3. At the nodes of Ranvier, lactate is exported by astrocytes via MCT1 or MCT4 and imported by the axon via MCT2.

**Figure 2 life-11-00238-f002:**
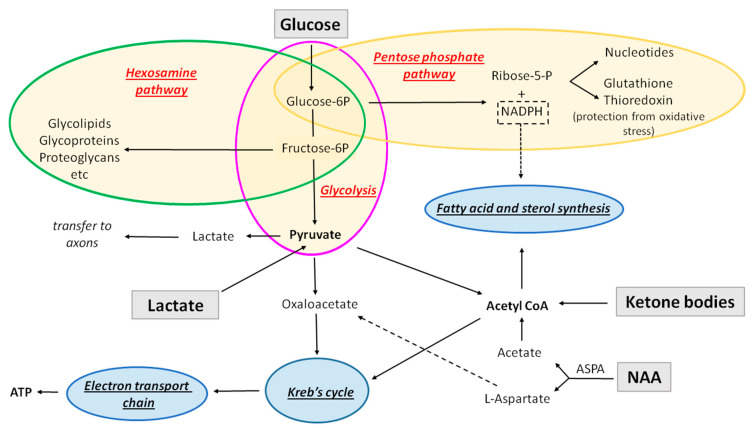
Schematic representation of metabolic pathways in oligodendroglial cells. Those indicated in red (glycolysis, pentose phosphate pathway, and hexosamine pathway) require glucose. The pathways circled in blue (fatty acid synthesis, Kreb’s cycle, and oxidative phosphorylation ( electron transport chain)) can also be fueled by alternative sources of energy, namely lactate, ketone bodies, and N-acetyl aspartate (NAA).

**Figure 4 life-11-00238-f004:**
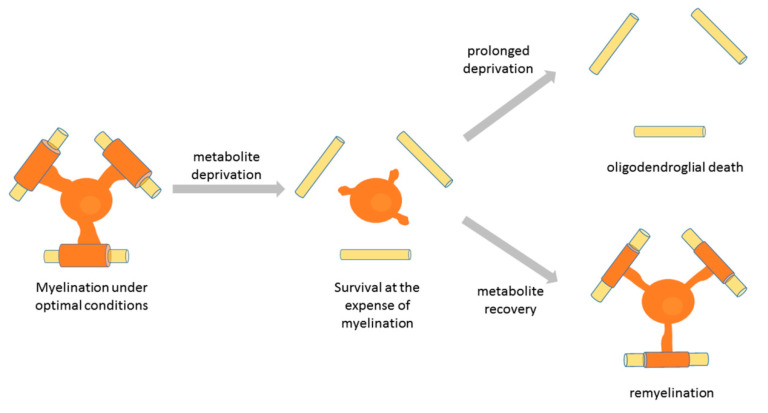
Response of myelinating oligodendrocytes to metabolic stress as a potential mechanism underlying dying-back oligodendropathy. Upon metabolic stress (nutrient deprivation), myelinating oligodendrocytes maintain a low glycolytic rate and upregulate autophagy, which enhances their survival, but results into process retraction and demyelination. If the stress does not prolong in time and metabolite supply is restored, oligodendrocytes can recover their normal metabolic rate, re-extend the processes, and remyelinate the axons. However, if nutrient deprivation persists, oligodendrocytes eventually die.

## Data Availability

No new data were created or analyzed in this study. Data sharing is not applicable to this article.

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
