# Peer review of "Oligodendroglial Energy Metabolism and (re)Myelination"

_life, 2021, doi:10.3390/life11030238_

Round 1

Reviewer 1 Report

This is a well-written review on the subject of oligodendroglial energy metabolism and (re) Myelination. The authors suggested that dysregulation of energy homeostasis in oligodendroglia may be an important contributor to myelin dysfunction in pathological conditions like multiple sclerosis and Alzheimer's disease.

Major Comments: 

  1. The abstract need some improvement. Some of the statements in the abstract are not supported by the information in the main text. The review seems to emphasize the importance of energy metabolism in oligodendroglia in the context of CNS disorders and diseases in the abstract, but in the main text, the information of functional relevance of energy metabolism in diseases is lacking. The author should add more information on that subject and cite papers or rephrase the abstract at the beginning.
  2. In the abstract the author states that the review will “highlight differences dependent on the maturation stage of the cell”, but the in the main text, it is very hard to find the relevant information. The author should include a subtitle on this subject.
  3. It seems the focus of the review is on the metabolic and biosynthetic pathways in the oligodendroglia cells, what is the functional importance of energy flow in oligodendroglia cells? The author need to elaborate and add more information.
  4. In the main text part 6, what is the relationship between nutrient deprivation and CNS disorders? Does the nutrient deprivation cause the demyelination that leads to the CNS neurological diseases, or the neurological conditions cause nutrient deprivation?
  5. Some the metabolic biosynthetic pathways are general and exist in all cell types, could the author emphasize the oligodendroglia cell-specific pathways?
  6. The figures need some major revisions. In figure 1, the diagram focus on the metabolite shuttling, why the shutting is vital to the myelination and re-myelination? Could the author explain what is metabolite shuttling and why it is important to the oligodendroglia function, myelination and re-myelination?
  7. The figure 2 is not well organized. Could the author realign the different pathways in an orderly fashion and use some regular shapes like circle or rectangle to represent the different pathways?

Minor comments:

  1. In figure 3, could the author emphasize the difference between the OPCs in myelination and myelin regeneration states?
  2. Could the author discus a bit more on the energy metabolism in the PNS oligodendroglia cells.
  3. In the introduction, the author should provide the full term for “PNS” when it appears for the first time.
  4. Some of keywords for the review are not important.

Author Response

Please, see attachment

Reviewer 2 Report

The authors have extensively described the mechanisms underlying oligodendroglial energy metabolism and have highlighted the differences dependent on the maturation stage of the cell.

The references are sufficient and succinct. Please check the entire document for minor spell check and grammatical errors. 

Author Response

Please, see attachment

Reviewer 3 Report

Tepavcevic summarized in this manuscript regarding on oligodendroglial energy metabolism and (re)myelination. The concept of this manuscript is basically very interesting. However, unfortunately, section 2.1 (2.1.1 lipid synthesis and 2.1.2 protein synthesis) is too short to summarize myelination. The author should consider to improve this section.

Author Response

Please, see attachment

Round 2

Reviewer 1 Report

The authors address most of the concerns.

Reviewer 3 Report

No comments.